# Sight for Sorghums: Comparisons of Satellite- and Ground-Based Sorghum Yield Estimates in Mali

**David B. Lobell [1,*], Stefania Di Tommaso [1], Calum You [1], Ismael Yacoubou Djima [2], Marshall Burke [1] and Talip Kilic [2]**

[1] Department of Earth System Science and Center on Food Security and the Environment, Stanford University, Stanford, CA 94305, USA; sditom@stanford.edu (S.D.T.); zcyou@alumni.stanford.edu (C.Y.); mburke@stanford.edu (M.B.)

[2] Living Standards Measurement Study (LSMS), Data Production and Methods Unit, Development Data Group, the World Bank, Washington, DC 20433, USA; iyacouboudjima@worldbank.org (I.Y.D.); tkilic@worldbank.org (T.K.)

* Correspondence: dlobell@stanford.edu

**Abstract:** The advent of multiple satellite systems capable of resolving smallholder agricultural plots raises possibilities for significant advances in measuring and understanding agricultural productivity in smallholder systems. However, since only imperfect yield data are typically available for model training and validation, assessing the accuracy of satellite-based estimates remains a central challenge. Leveraging a survey experiment in Mali, this study uses plot-level sorghum yield estimates, based on farmer reporting and crop cutting, to construct and evaluate estimates from three satellite-based sensors. Consistent with prior work, the analysis indicates low correlation between the ground-based yield measures (r = 0.33). Satellite greenness, as measured by the growing season peak value of the green chlorophyll vegetation index from Sentinel-2, correlates much more strongly with crop cut (r = 0.48) than with self-reported (r = 0.22) yields. Given the inevitable limitations of ground-based measures, the paper reports the results from the regressions of self-reported, crop cut, and (crop cut-calibrated) satellite sorghum yields. The regression covariates explain more than twice as much variation in calibrated satellite yields ($R^2$ = 0.25) compared to self-reported or crop cut yields, suggesting that a satellite-based approach anchored in crop cuts can be used to track sorghum yields as well or perhaps better than traditional measures. Finally, the paper gauges the sensitivity of yield predictions to the use of Sentinel-2 versus higher-resolution imagery from Planetscope and DigitalGlobe. All three sensors exhibit similar performance, suggesting little gains from finer resolutions in this system.

**Keywords:** crop yield estimation; crop cutting; remote sensing; Mali; sub-Saharan Africa

## 1. Introduction

Agriculture remains a key contributor to national employment and economic output in many countries throughout the world. Improving agricultural productivity is thus a key engine for both increasing local food security as well as spurring overall economic growth [1]. Finding policy solutions that can improve performance depends on understanding the drivers of agricultural productivity, which in turn depends on the ability to accurately measure productivity.

A widely used measure of cropland productivity is crop yield, defined as the weight of edible product (e.g., kilogram of grain) produced per unit area (e.g., hectare). Although yield alone cannot capture all aspects of household income or well-being, it is a key determinant of the profitability of

farm management as well as household food security [2,3], and is among the most commonly used indicators to gauge the status or rate of progress of agricultural systems [4].

Despite its simplicity relative to other measures of productivity, accurately measuring crop yields presents a challenge in smallholder agricultural systems, where plots are typically less than 2 ha in size. Past research has primarily relied on either farmer self-reports or objective crop cuts to measure yields. In the former, survey enumerators ask farmers a series of questions related to total grain harvest and cultivated area, and compute yields as the ratio of reported production to reported area. Among the many sources of errors in these estimates are the frequent use of non-standard measurement units by farmers (which then require conversion to a common unit), the various moisture levels at which grain is harvested, partial harvest before crop maturity for home consumption (e.g., green maize), a tendency to round off numbers, and unobserved incentives to farmers to make their land seem more or less productive [5,6]. Self-reported yields are particularly problematic in subsistence systems where most food is grown for home consumption and where accurate record keeping by farmers is not the norm. Recent research has also demonstrated systematic measurement errors in self-reported yields, with a direct bearing on the conclusions of economic research [6–8]. Nonetheless, most national surveys, such as those supported by the World Bank's Living Standards Measurement Study-Integrated Surveys on Agriculture (LSMS-ISA), still permit the computation of self-reported yields.

The second common way to study productivity, typically referred to as a crop cut, is to measure the grain weight harvested from a randomly selected portion of a farmers' plot [9]. Assuming that the crop cut locations are truly random and avoid preferential selection of the middle or edges of the plot, they have been shown to provide an unbiased estimate of the true yield based on full-plot harvests [6]. A large number of crop cuts conducted in a region can therefore give a reliable measure of average yields. However, errors for individual plots can still be large, since yields can exhibit considerable spatial heterogeneity within plots and even relatively large crop cut areas, such as two 50–75 m$^2$ plots per plot suggested in [10], will represent only a small fraction of the total plot area. More commonly, much smaller crop cut areas are used, such as three 2 m$^2$ sub-plots in Jain et al. [11] or three 4 m$^2$ quadrants in Lambert et al. [12].

Satellite remote sensing offers a potential alternative approach to measuring crop yields, especially as satellite sensors with the fine spatial resolution needed to distinguish individual smallholder plots become more prevalent. Several decades of research has focused on developing and testing algorithms to estimate yields from satellite, initially in large commercial plots [13,14] and increasingly in smallholder systems [11,12,15,16]. These studies have primarily focused on the primary staple crops within a region, such as maize, wheat, and rice, although some recent work has extended beyond, such as to cotton, millet, and sorghum [12].

In the current study, we assess the accuracy of satellite-based estimates of sorghum yields in the main sorghum growing region of Mali. Similar to many sub-Saharan African countries, Mali's agricultural sector comprises a major portion of the GDP and employment, with respective shares of 39% (as measured by the percent value added GDP for agriculture, forestry, and fishing) and 65% in 2018 (retrieved from data.worldbank.org). In terms of cultivated area by smallholder farmers, cereal production is 77% of the total crop production, of which 26% is sorghum, making it an important staple crop for Mali's 17 million people (authors' calculation based on the 2017 Mali LSMS-ISA data).

This work contributes to the literature on remote sensing of yields in three primary ways. First, we focus on sorghum, one of the primary staples in sub-Saharan Africa but less commonly studied compared to other staples such as maize and rice. Sorghum is a relatively difficult crop for remote yield estimation given the high variability within and between plots in the cultivars grown by farmers, and the relatively high variation in the harvest index (ratio of grain to total crop biomass) compared to other crops [12].

Second, we systematically compare satellite measures to both self-reports and crop cuts, and across a much larger geographic domain and number of plots than is typically done. For example, the

Lambert et al. [12] study in Mali measured yields on 27 total sorghum plots, whereas the current study covers 575 plots.

Third, we compare estimates from publicly available Sentinel-2 imagery (10 m spatial resolution) with estimates from very high resolution (VHR) Planetscope (~3 m resolution) and DigitalGlobe multispectral imagery (~1–2 m resolution) to assess the added value of VHR for this application. Given the significantly higher costs of acquisition of VHR imagery, it is informative for budget-constrained researchers and development organizations to assess the relationship, if any, between image cost and accuracy of agricultural productivity measures.

The following section describes the data used in this study. Section 3 then describes the methods used to evaluate the quality of both the ground-based and satellite-based yield estimates. Section 4 then discusses the results, while Section 5 summarizes the study's conclusions.

## 2. Data

### 2.1. Survey and Crop Cut Data

Our study region was the Dioïla Cercle, an administrative subdivision in the southeastern part of the Koulikoro region of Mali (Figure 1A). This area is within the primary sorghum-producing area in Mali and has been the locus of many activities by the International Crops Research Institute for the Semi-Arid Tropics (ICRISAT). Key to our research is a farm survey that was implemented during the 2017 agricultural season. The survey fieldwork was conducted from August 2017 to February 2018 by ICRISAT-Mali, under the supervision of and technical assistance from the World Bank Living Standards Measurement Study (LSMS) team. Within Dioïla, four 10 × 10 km blocks were identified to ensure heterogeneity of topographic relief (Figure 1B). In each block, the complete list of villages was constructed based on the shapefiles from the 2009 Population and Housing Census. Overall, there were 17 villages across all blocks. In each village, a complete listing of households was carried out, as part of which the sorghum-producing households with at least one purestand plot were identified. Given the low incidence of intercropped sorghum plots in Dioïla (estimated at 6% according to the 2017 LSMS-ISA survey), we elected to exclusively sample purestand plots. Subsequently, we randomly selected 150 of these households in each block. The across-village allocation of the sampled households in each block was proportional to the village-level total count of households with purestand sorghum plots.

Each sampled household was in turn visited three times. The first visit's fieldwork spanned the period of mid-September–October 2017. During the first visit, the post-planting questionnaire collected detailed information on household composition and demographics, housing conditions, ownership of consumer durables and agricultural implements, receipt of agricultural extension services, and farm organization. The questionnaire modules regarding the latter collected farmer-reported information for (i) all parcels owned and/or cultivated by the household and (ii) all plots cultivated with sorghum within these parcels, on a variety of topics, including parcel and plot areas, parcel ownership and tenure, plot management and farming practices, plot-level labor inputs for land preparation and planting, and plot-level sorghum cultivation and varietal attributes.

One purestand sorghum plot was in turn selected at random in each household. The area and boundaries for the selected sorghum plot were captured via a handheld Garmin eTrex 30 GPS unit with the Wide Area Augmentation System enabled for higher accuracy. An 8 × 8 m crop cut sub-plot was then established on each plot for the actual processing and weighing the crop harvest at the end of the season. The latitude and longitude of the edges of each sub-plot were recorded by Android tablets, and each sub-plot was sub-divided into four 4 × 4 m quadrants, for which the crop harvest was processed and weighed separately. The approach of a random placement of the crop cut sub-plots, supervision of the crop cut sub-plots throughout the season, and harvesting, processing, and tracking of sorghum cultivated in each quadrant were all identical to the approach in an earlier methodological study focused on maize in Eastern Uganda [6] (please see [6] and their Appendix D for more details).

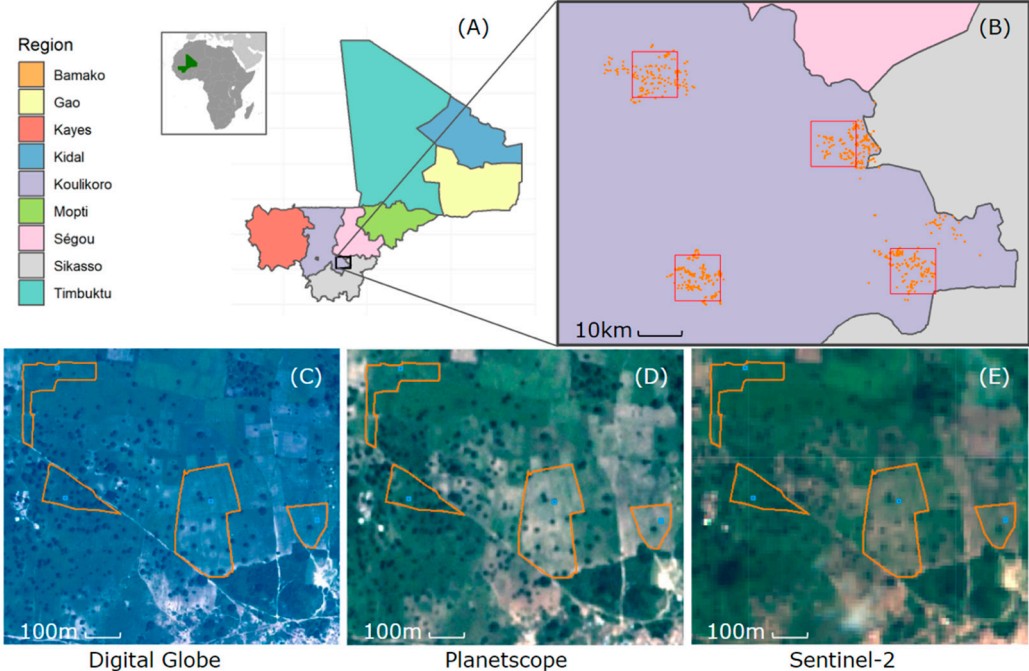

**Figure 1.** (**A**) Map of Mali with the study region outlined in black. (**B**) Study region in Koulikoro showing the four distinct survey sites. The location of captured DigitalGlobe images is shown in red. (**C**) Example of a DigitalGlobe image collected on 19 September 2017, with plot boundaries for a subset of plot locations shown in orange and the centroid of the crop cut sub-plot shown in blue. (**D**–**E**) Same as (**C**) but background image is a (**D**) Planetscope image from 11 September 2017 and a (**E**) Sentinel-2 image from 13 September 2017.

The second visit's fieldwork was conducted during the harvest season, over the period of November–December 2017. During the second visit, the harvest from each crop cut quadrant was barcoded and processed for drying, which was done at a centralized location in each village, under the supervision of the resident enumerator. Finally, the third visit fieldwork spanned the period of late-January–February 2018. During the third visit, the dried harvest associated with each crop cut quadrant was weighed, and the households were administered a post-harvest questionnaire that elicited farmer-reported information for all plots cultivated with sorghum regarding (i) sorghum production (allowing for the use of non-standard measurement units for the quantification of production), (ii) use of non-labor inputs, including organic and inorganic fertilizers, as well as pesticides, and (iii) household and hired labor inputs for weeding, input application, and crop harvest.

The final dataset consisted of household- and plot-level information augmented with self-reported and GPS-based plot areas, self-reported and crop cut sorghum harvest weights, and plot boundary polygons. GPS plot boundaries were processed to remove any self-intersections and to erase any areas of overlap between neighboring plot polygons. Crop cut yields were calculated by dividing the total grain weight of sorghum by the 64 m$^2$ designated crop cut area. Grain moisture was deemed to be uniformly low, given the arid climate of the region, and was not measured or adjusted for in the crop cut yield estimates. A total of 25 fields were omitted from subsequent analysis, either because of missing crop cut yields or self-intersecting GPS boundaries, leaving a final sample size of 575 plots.

Self-report yields were calculated by first converting each survey answer to kilograms (kg), since responses were provided in varying non-standard measurement units such as a "sheaf". To obtain kg-equivalent production measures, village-level median conversion factors were computed based on the conversion factors that were provided by the respondents for each non-standard unit during the third interview that solicited self-reported information on agricultural production. (We compared our conversion factors to those that were obtained from a pilot study that was conducted by the Ministry

of Agriculture (MoA), with support from the World Bank LSMS-ISA. The conversion factors matched across the two studies, even though "bunches" exhibited a large variance. The scaled-up version of the MoA conversion factor survey is slotted for implementation starting in December 2019.) Further, if the reported harvest condition was unshelled, a shelling factor was applied. We computed the conversion factors at the village level in part to reduce potential biases in farmer-reported conversions, while recognizing the possibility of spatial variation in conversion factors based on prior work [17]. In the current study, only 4% of observed variability in self-report yields were explained by the conversion factors used.

The total harvest weights in kg were then divided by the self-reported plot area to obtain self-reported yields. In sensitivity tests, self-reported yields were also calculated using the polygon area, with any notable differences discussed below. The preference for calculating self-reported yields using self-reported area is based on Abay et al. [7], who show that if measurement errors in self-reported production and area are correlated, correcting one (area) can lead to bigger errors in analyzing yields than correcting neither.

## 2.2. Satellite Data

We used Sentinel-2 as the primary source of remote sensing imagery for this study. The Sentinel-2 Multispectral Instrument (MSI) measures 13 spectral bands that collectively span the visible/near infrared and short wave infrared spectral range, from roughly 440–2200 nm. For the current study we used the Sentinel-2 Level 1-C product, which represents top-of-atmosphere reflectance. From these raw bands we calculated several common vegetation indices that have been found to be useful for agricultural monitoring in similar settings [12,15,16].

Specifically, we computed the Green Chlorophyll Vegetation Index (GCVI) [18] at 10 m resolution, the Normalised Difference Vegetation Index (NDVI) [19] at 10 m resolution, and the MERIS Terrestrial Chlorophyll Index (MTCI) [20] at 20 m resolution:

$$\text{GCVI} = (R_{842}/R_{560}) - 1 = (B8/B3) - 1, \tag{1}$$

$$\text{NDVI} = (R_{842} - R_{665})/(R_{842} + R_{665}) = (B8 - B4)/(B8 + B4), \tag{2}$$

$$\text{MTCI} = (R_{842} - R_{705})/(R_{842} - R_{665}) = (B8 - B5)/(B8 - B4), \tag{3}$$

where $R_\lambda$ refers to reflectance centered at wavelength $\lambda$ and B refers to the corresponding Sentinel-2 band. Sentinel-2 has a revisit period of 5 days in this study region. Based on the reported sowing and harvest dates from the household survey, we used the time series from 5 May 2017 through to 31 December 2017, for a total of 87 unique image dates for each plot.

For higher-resolution imagery, we considered two alternatives. First, DigitalGlobe (DG) images were acquired by their GeoEye-1 (1.84 m multispectral resolution), WorldView-2 (1.84 m), and WorldView-3 (1.24 m) sensors on a monthly basis from July through December 2017 for the four 10 × 10 km (red) blocks shown in Figure 1. Given the limited swath of DG images, only a fraction of plots (typically less than one-quarter) were covered by any individual image. Removing images with excessive cloud cover resulted in successful acquisition of 18 images, with the dates for each image shown in Table 1.

**Table 1.** Summary of images acquired by DigitalGlobe (DG) sensors in the study region. Region refers to the quadrant shown in Figure 1. Sensor source is shown in parentheses G = GeoEye-1, W2 = WorldView-2, and W3 = WorldView-3.

| Region | Digital Globe Imagery Dates |
|--------|------------------------------|
| 1 | 8/17/17 (G), 9/19/17 (G), 10/1/17 (W2), 10/18/17 (W3), 11/7/17 (G) |
| 2 | 9/5/17 (G), 10/16/17 (G), 11/4/17 (G) |
| 3 | 9/5/17 (W3), 10/5/17 (G), 10/16/17 (G), 11/8/17 (W2) |
| 4 | 8/1/17 (G), 8/29/17 (W3), 9/19/17 (G), 10/6/17 (W2), 10/16/17 (G), 11/6/17 (W3) |

Second, approximately 3 m resolution Planetscope images were collected at roughly daily frequency by sensors on the Planet company's "dove" of Planetscope cubesats. The Planet's constellation consists of >100 small cubesats in low-Earth orbit. In contrast to the large size of traditional VHR sensors (e.g., WorldView-3 weighs 2800 kg), Planet doves weigh approximately 5 kg and thus have dramatically lower launch costs. Although the image quality is lower in terms of both sensor signal-to-noise ratio and spatial resolution, the large number of doves allows frequent observations at any point on the Earth's surface. For the current study, access to Planetscope images was provided via Planet's ambassador program. We downloaded all images in the study region with fewer than 5% of clouds, with an average of 52 observations per plot throughout the growing season. All images were converted to top-of-atmosphere reflectance using coefficients provided in the image metadata.

## 3. Methods

### 3.1. Satellite Data Processing

For each of the 575 plots, Sentinel-2-based vegetation index values were computed for both the entire plot and for the $8 \times 8$ m crop cut sub-plot on each plot. A common challenge with optical satellite imagery is the prevalence of clouds that obscure or completely block the satellite sensor's view of the land surface. Sentinel-2 has a predefined cloud mask band, but preliminary analysis revealed this mask was unreliable in the region—with many false positives and false negatives. Rather than explicitly flagging cloudy observations, we adopted a recursive curve fitting procedure similar to that implemented in the Timesat software package [21], which is robust to the inclusion of cloudy observations.

Specifically, a discrete Fourier transform, also known as a "harmonic regression," was fit to all 87 observations in a pixel between 5 May and 31 December:

$$f(t) = c + \sum_{k=1}^{N} (a_k \cos 2k\omega\pi t + b_k \sin 2k\omega\pi t), \tag{4}$$

where $f(t)$ is the value of the reflectance band or vegetation index of interest, $t$ is the date of observation, N is the number of harmonic terms included, and $a_k$, $b_k$, and, $c$ are cosine, sine, and intercept coefficients estimated by the regression, respectively. Past research has shown that harmonic regressions can adequately characterize vegetation phenology, including in agricultural settings [21–23]. Here we used the second-order harmonic ($N = 2$) following [22].

When fit to the raw data, Equation (4) is heavily influenced by cloudy observations, as illustrated in Figure 2 for the time series of GCVI at a representative plot. The raw values shown in black dots display frequent drops in value to near zero, indicating cloud cover on those days. Therefore, an iterative procedure was used whereby the predicted values from Equation (4) were compared with the input value at each date. If the input value was lower, it was replaced with the predicted value, and the resulting time series was then used to refit Equation (4). This process can be repeated until the influence of cloudy observations is minimized. In this case, we found that the 10th iteration approximated the

upper limits of the curve more closely, as shown in Figure 2. Thus, for all plots we used the fitted values from the 10th iteration in further analysis.

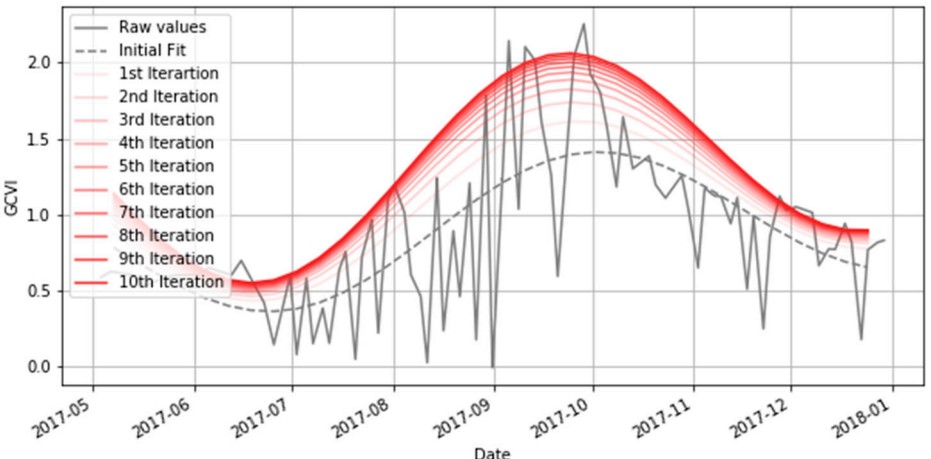

**Figure 2.** Example of a recursive harmonic fit applied to a typical plot time series.

### 3.2. Calibration of Satellite-Based Yield Model

To convert satellite vegetation indices to yield, we performed a simple regression between the peak value of the harmonic curve (VI_peak) on each plot *i*, and the associated ground-based yield measure (Y):

$$Y_i = \alpha + \beta_1 VI_{peak,i} + \varepsilon_i \,. \tag{5}$$

When crop cut yields were used for Equation (5), the VI was taken from the Sentinel-2 pixel that contained the centroid of the 8 × 8 crop cut sub-plot. Taking an average of pixels around this point was also tried but had little effect on the results. When self-reported yields were used, the VI for each plot was computed as an average of values from all non-tree pixels with at least half of their area falling within the plot boundary. Following [12], we masked out pixels with trees since plots often contain enough trees to appreciably influence the apparent greenness of the plot. In our case, pixels with trees were identified as those with a GCVI harmonic fit that exceeded a value of 1.0 at any point between 1 June and 15 July, which is the beginning of the crop growing season when crop pixels were typically well below this value (Figure 2). Overall, 34% of the pixels within plots were removed in this manner.

We alternatively tested GCVI, NDVI, and MTCI as potential predictors of yield, with GCVI generally performing best. Models with additional terms, such as the date on which the peak occurred or the fitted value of the harmonic at earlier times in the season, were also tested but did not significantly improve the model and therefore are not presented for the sake of brevity.

### 3.3. Comparison of Survey and Satellite Data

A major difficulty in assessing the quality of satellite yield estimates is the fact that the traditional ground-based measures (i.e., self-reported and crop cut yields) are themselves prone to errors as discussed in the introduction. The most common approach has simply been to treat ground data as "ground truth," and attribute any differences between satellite and ground-estimates to errors in the satellite data. In this case, the calibration $R^2$ of Equation (5) is a common metric of how well the satellite-based yield model performs. An alternative to directly comparing the two measures is to use a vector of variables (*X*) that are expected to influence yields (*Y*), and test whether the regression

$$Y_i = \alpha + \beta X_i + \varepsilon_i \tag{6}$$

results in weaker, stronger, or equivalent $\beta$ coefficients (with expected signs) when using satellite-based yields. This approach avoids dependence on the assumption that the ground-based measures are free of error, and was used to evaluate estimates of maize yields in recent studies in Kenya [15] and Uganda [16]. Similar to the prior work, the following (self-reported) variables are included in our vector *X*: Household size; head of household age, and education level; plot size; plot distance to (i) household, (ii) road, and (iii) market; use of crop rotation and fallowing; frequency of ploughing; quantity of labor, seed, fertilizer, and pesticides used; and rainfall totals for the entire growing season and for the month of August, which is especially important for determining drought stress during flowering. Rainfall was obtained for each 5-day period in the growing season from the CHIRPS dataset [24]. The coefficients and overall explanatory power of the regressions were then compared for models that alternatively used self-reported, crop cut, or satellite-based yields.

## 4. Results and Discussion

Summary statistics for survey responses and crop cut yields are presented in Table 2. Overall, farmers reported an average plot size of 1.88 hectares (ha), which is 23% larger than the 1.53 ha average size of the plot polygons measured with GPS. This tendency to overestimate plot area was particularly noteworthy on plots below 2 ha, consistent with the findings in other LSMS-ISA studies where plot sizes were overestimated by farmers on plots below 2 ha in Malawi, Uganda, Tanzania, and Niger [25].

**Table 2.** Summary statistics on selected variables, which are either farmer-reported or measured by field staff. Three fields lacked self-reported yields, and six lacked reported harvest dates.

| Continuous Variables | Source | *n* | Min | Median | Max | Mean |
|---|---|---|---|---|---|---|
| Longitude | Measured | 575 | −6.80 | −6.35 | −6.03 | −6.44 |
| Latitude | Measured | 575 | 12.13 | −12.30 | 12.65 | 12.35 |
| Elevation | Measured | 575 | 234.2 | 300.8 | 370.5 | 302.6 |
| Yield—crop cut (kg/ha) | Measured | 575 | 0 | 421.7 | 2472.5 | 497.5 |
| Yield—survey (kg/ha) | Reported | 572 | 0 | 319.6 | 6189.4 | 475.6 |
| GPS plot area (ha) | Measured | 575 | 0.05 | 1.18 | 11.93 | 1.53 |
| Farmer plot area (ha) | Reported | 575 | 0.10 | 1.50 | 12.00 | 1.88 |
| Sowing date (half-month) | Reported | 575 | April 2nd half | June 2nd half | August 2nd half | - |
| Harvest date | Measured | 569 | 28 October 2017 | 12 November 2017 | 8 December 2017 | - |
| Distance to household (km) | Reported | 575 | 0 | 2 | 20 | 2.29 |
| Distance to road (km) | Reported | 575 | 0 | 1 | 21 | 2.16 |
| Distance to market (km) | Reported | 575 | 0 | 7 | 30 | 7.52 |
| Labor for sowing (h/ha) | Reported | 575 | 2.4 | 31.0 | 2521.5 | 81.8 |
| Labor for harvest (h/ha) | Reported | 575 | 0.7 | 30.0 | 1011.3 | 44.8 |
| Inorganic fertilizer (kg/ha) | Reported | 575 | 0 | 0 | 300 | 19.7 |
| Seed used (kg/ha) | Reported | 575 | 3.6 | 5.0 | 20.0 | 5.8 |
| **Binary Variables** | **Source** | ***n*** | **Yes** | **No** | **Mean** | |
| Fallowed in the past 10 years? | Reported | 575 | 34 | 541 | 0.059 | |
| Practices crop rotation? | Reported | 575 | 482 | 93 | 0.84 | |
| Field has erosion problems? | Reported | 575 | 115 | 460 | 0.2 | |
| **Categorical Variables** | **Source** | ***n*** | **Top Counts** | | | |
| Head of HH—Ethnicity | Reported | 575 | Bambara/Malinké: 338, Peulh/Foulfoulbé: 188 | | | |
| Soil type | Reported | 575 | Clay: 322, Silt-Sand: 97, Sand-Silt: 93 | | | |
| Soil fertility | Reported | 575 | Low: 106, Average: 301, Good: 168 | | | |
| Terrain type | Reported | 575 | Plain: 426, Gentle Slope: 95, Steep Slope: 41 | | | |
| Tree coverage | Reported | 575 | None: 29, Few: 376, Many: 170 | | | |

Only 31% of farmers reported using any inorganic fertilizer, with a mean reported rate of just below 20 kg/ha. The majority of farmers reported having clay soils and a plain (flat) terrain. Although fewer than 10% of farmers reported having a steeply sloping plot, 20% reported having had erosion problems.

Both self-reported and crop cut yields exhibited similar mean values of just below 500 kg/ha. However, the median yield was roughly 25% lower for self-reports, consistent with the overestimation of areas by a similar amount. At the same time, self-reports had a much higher upper bound of yields stretching to values above 6000 kg/ha, whereas crop cuts never exceeded 2500 kg/ha. This is striking given that the crop cuts cover a much smaller area (64 $m^2$ is 0.4% of the mean plot size of 15,303 $m^2$), and because of sub-plot heterogeneity they would be expected to exhibit more extreme values than self-reports that correspond to the plot-level mean yield.

A scatter plot between the self-report and crop cut yields on each plot reveals a striking lack of agreement between the two ground-based measures (Figure 3). The overall correlation between the two measures was just 0.33, indicating that one ground-based measure can explain only approximately 11% of variation in the other. The correlation slightly improves to 0.40 when omitting plots with self-report values above 2500 kg/ha (Figure 3B). The low agreement between ground-based measures is similar to that reported for maize plots in Eastern Uganda, where self-report and crop cut yields had a correlation below 0.30 even after excluding very small plots where self-reports were deemed least reliable [16].

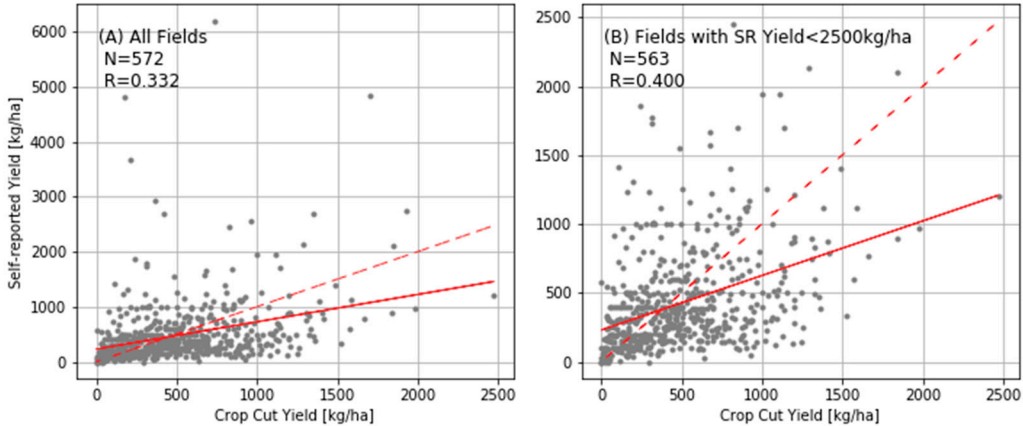

**Figure 3.** Comparison of crop cut vs. self-reported yields for (**A**) all fields, (**B**) removing SR yields above 2500 kg/ha. Best-fit regression line is shown in solid red, 1:1 line shown in dashed red. In general, the two yield measures show low agreement ($R^2 < 0.2$) even when removing apparent outliers in self-reported yields.

Moving to the comparison with satellite measures, Figure 4 shows the correlation between the two ground-based yield measures and GCVI for each date of Sentinel-2 imagery, both for the raw GCVI values and the harmonic fit at those dates. We find the highest correlations between GCVI and ground-based yields were observed in mid- to late-September for both self-report and crop cut yields, which is generally the same time that the GCVI curve reaches its maximum (Figure 2). The GCVI correlations with crop cut yields were consistently higher than those with self-report yields, with a peak value of 0.48 for crop cut compared to 0.23 for self-reports (Figure 4). The self-report correlation improved slightly to 0.27 when removing fields with self-report yields above 2500 kg/ha, but still remained well below the corresponding correlation for crop cuts. This finding suggests that the self-report yields are the less reliable of the two ground measures.

The decision to use $VI_{peak}$ in Equation (5) is supported by the observation that the GCVI-yield correlation peaks in late September (Figure 4). Results were similar when using NDVI, with r = 0.48 for GCVI as compared to r = 0.50 for NDVI. MTCI was found to be less suitable to the harmonic fitting because clouds did not systematically reduce MTCI values; nonetheless MTCI performed similarly

to GCVI and NDVI for clear images in this setting, with a peak correlation of 0.40 compared to 0.45 for GCVI.

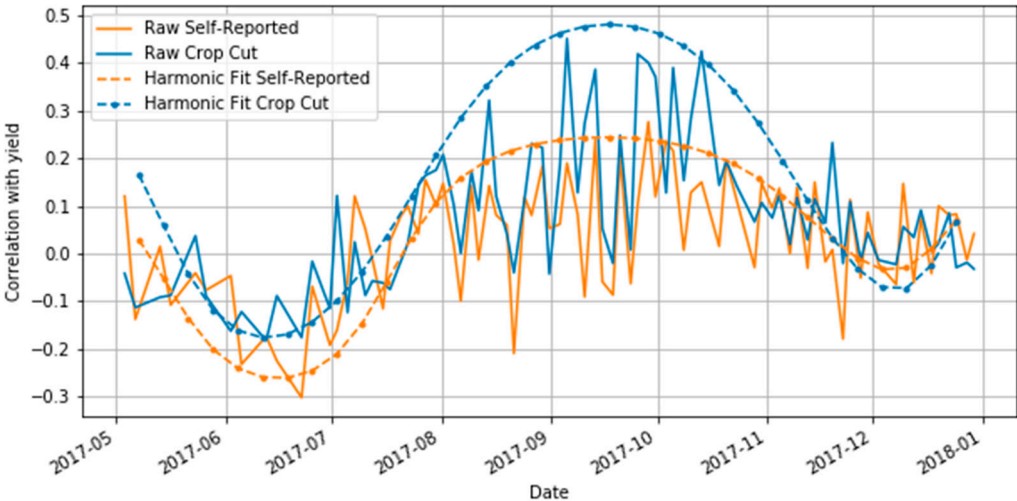

**Figure 4.** Correlation of yield and GCVI by date. Weekly max values are the maximum correlation in that week against the raw due to instances of multiple images; harmonic fits are the predicted value from the 10th recursive iteration. Crop cut time series show higher correlations and both peak in late September, around a month before harvest.

Predictions from Equation (5) using GCVI were significantly correlated with both ground-based measures (Figure 5, $p < 0.01$ for slopes in both panels), with better agreement with crop cut than self-report yields. Although GCVI successfully captured some of the variation in ground-based yield measures, well over half of the variation was not captured. At least in part, these discrepancies arise because of noise in the ground-based measures, namely reporting errors in the case of self-reports or the effects of spatial heterogeneity in the case of crop cuts.

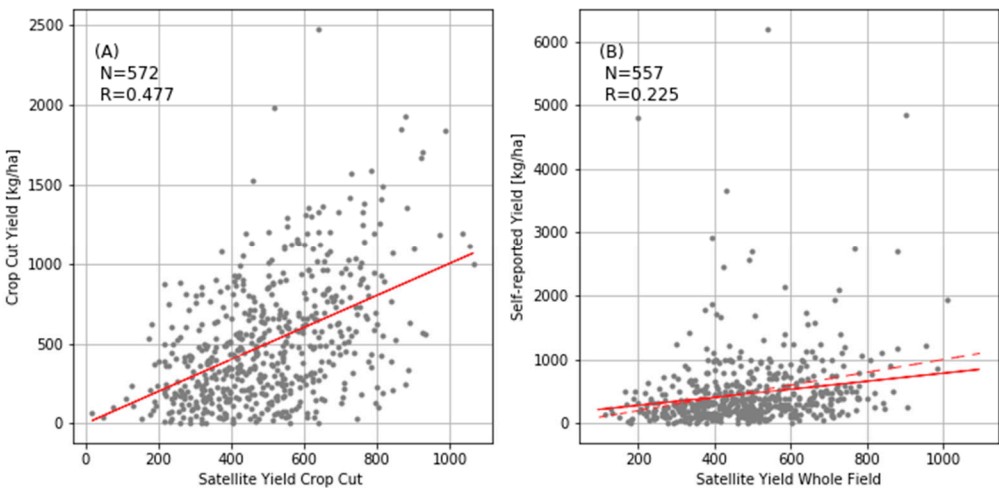

**Figure 5.** Comparison of satellite-based yield estimates to the ground-data used to calibrate the model. (**A**) crop cut yields and (**B**) self-reported yields. Best-fit regression line is shown in solid red, 1:1 line shown in dashed red. Both satellite-based models used peak GCVI from the 10th harmonic regression, as illustrated in Figure 2. In general, GCVI was more strongly correlated with crop cut than with self-reported yields.



As a separate assessment, we performed the regression in Equation (6) using three alternative measures of yield: Self-report, crop cut, and satellite-based estimates using the model calibrated to crop cuts. For predicting field-scale yields with GCVI, we use the mean value of GCVI for the entire plot (again excluding pixels with trees), rather than for the crop cut sub-plot used in the calibration step. Table 3 summarizes the coefficients and explanatory power of each model. All models were statistically significant, although less than one-quarter of total variation in measured yields was captured in all cases. The models' explanatory power was likely limited by a lack of objective soil measures in the current study, especially given overall low levels of input use in the region (Table 2). Consistent with a high importance of soil, all models showed a statistically significant ($p < 0.01$) negative association between yields and whether the plot had been fallowed in the past decade (a likely indication of poor soil quality).

**Table 3.** Regression results using three alternative measures of sorghum yield: (i) Self-reported, (ii) crop cut, and (iii) crop cut-calibrated Sentinel-2 satellite yields. In first column, carrots (ˆ) indicate inputs that were input as a binary variable (0,1) and asterisk (*) indicate variables that were calculated on a per ha basis using self-report area. Values in parentheses show the standard error of the estimate, and significance of the coefficient is indicated as (* $p < 0.1$, ** $p < 0.05$, *** $p < 0.01$).

| | Log Self-Report Yield * | Log Crop-Cut Yield | Log Satellite Yield |
|---|---|---|---|
| Constant | −7.22 (14.09) | −2.08 (6.66) | 6.77 ** (3.26) |
| Log Plot Area (ha) * | −0.15 ** (0.07) | 0.08 (0.07) | −0.02 (0.02) |
| Household Size (persons) | 0.01 ** (0.0) | 0.0 (0.0) | 0.0 (0.0) |
| HoH Age (yrs) | 0.0 (0.0) | −0.0 (0.0) | 0.0 (0.0) |
| HoH Education (yrs) | −0.03 * (0.02) | −0.03 * (0.02) | −0.02 *** (0.01) |
| Distance to Household (km) | −0.02 (0.02) | 0.01 (0.02) | 0.02 ** (0.01) |
| Distance to Road (km) | 0.01 (0.02) | 0.01 (0.01) | −0.0 (0.01) |
| Distance to Market (km) | 0.01 (0.01) | 0.01 (0.01) | 0.0 (0.0) |
| Practices Crop Rotation ˆ | 0.16 (0.11) | 0.28 *** (0.1) | 0.05 (0.04) |
| Fallowed < 10 Years Ago ˆ | −0.27 *** (0.09) | −0.6 *** (0.17) | −0.24 *** (0.08) |
| Erosion Problems ˆ | −0.08 (0.09) | −0.04 (0.1) | −0.04 (0.03) |
| Ploughs (times) | 0.02 (0.04) | 0.03 (0.04) | 0.05 (0.03) |
| Sowing Labour * (hrs/ha) | 0.0 (0.0) | 0.0 * (0.0) | 0 * (0) |
| Harvest Labour * (hrs/ha) | 0.0 ** (0.0) | −0.0 (0.0) | −0.0 (0.0) |
| Log Seed Quantity * (kg/ha) | 0.15 (0.18) | −0.12 (0.12) | 0.02 (0.05) |
| Inorganic Fertiliser * (kg/ha) | 0.0 * (0.0) | 0.0 (0.0) | 0.0 (0.0) |
| Pesticide * (kg/ha) | 0.0 (0.0) | 0.0 (0.0) | −0.0 (0.0) |
| Log Total Precipitation (mm) | −1.25 (1.88) | −0.8 (0.65) | −2.49 *** (0.39) |
| Log Aug Precipitation (mm) | 3.71 *** (1.25) | 2.36 (1.5) | 2.84 *** (0.46) |
| F-statistic | $-7.0 \times 10^{13}$ (df = 18;519) | $1.78 \times 10^{10}$ *** (df = 18;519) | $6.32 \times 10^{11}$ *** (df = 18;519) |
| Observations | 538 | 538 | 538 |
| R_squared | 0.14 | 0.08 | 0.26 |
| R_squared_adj | 0.11 | 0.05 | 0.24 |

The model using crop cut yields exhibited the lowest explanatory power (adjusted $R^2 = 0.05$), likely reflecting the fact that the crop cut sub-plot areas represent on average less than 1% of the total plot area that pertains to the survey responses on management and soil conditions. The model using self-report yields had higher explanatory power (adjusted $R^2 = 0.11$). Several factors likely contribute to this increase, including that (i) self-report yields (unlike crop cut yields) correspond to the same spatial scale as other self-reported plot characteristics; (ii) measurement errors for different survey questions are likely correlated and could inflate the overall model performance, for instance if a farmer who overestimates production tends to also overestimate seed or fertilizer inputs; (iii) farmer perceptions of yield are influenced by the weather during the growing season, as suggested by the fact that the association with August rainfall is larger for self-report yields than the other measures; and (iv) self-reported yields tend to be higher on smaller plots, as indicated by a significant negative coefficient for plot size. Although this last factor increases the explanatory power of the model, we interpret it as an artefact of self-report bias on small fields rather than a true inverse relationship between plot size and productivity, as discussed in [6].

The model using crop cut-calibrated satellite yields exhibited the highest explanatory power among the three models (adjusted $R^2$ = 0.24), more than twice as much as for crop cut or self-reported yields. One interpretation of this could be that errors in the satellite model are correlated with the some of the factors in $X$. For example, GCVI is known to correlate strongly with overall canopy biomass and nitrogen content [26,27], but the relationship between biomass and yield is variable in rainfed sorghum systems [12]. Therefore, it is possible that inputs such as fertilizer are more predictive of biomass than yield, while at the same time our satellite-based estimates are more sensitive to biomass than yield. Unfortunately, without explicit measures of biomass we cannot assess the extent to which this explanation holds. Another potential explanation, in our view a more likely one, is that the higher explanatory power indicates a greater ability of satellite data to detect plot-level variations in yield compared to either self-reports or crop cut estimates.

The results discussed above all pertain to satellite estimates obtained when using ground data from all 575 fields to calibrate Equation (5). A practical question is whether similar results could be achieved at lower cost by using significantly fewer plot samples. To test this, we varied the number of ground samples used in calibration from 5 to 500, each time randomly selecting the calibration samples and reserving the remaining fields as a validation set. Figure 6 shows the average root mean square error (RMSE) for the validation set, averaged over all validation points and for 200 different iterations with different randomly selected calibration samples. The out-of-sample performance improves most rapidly up to roughly 30 ground samples, after which RMSE exhibits a more gradual decrease. Also shown in Figure 6 is the RMSE when the model is calibrated to self-reported yields, which consistently do worse than crop cuts regardless of how many ground samples are obtained.

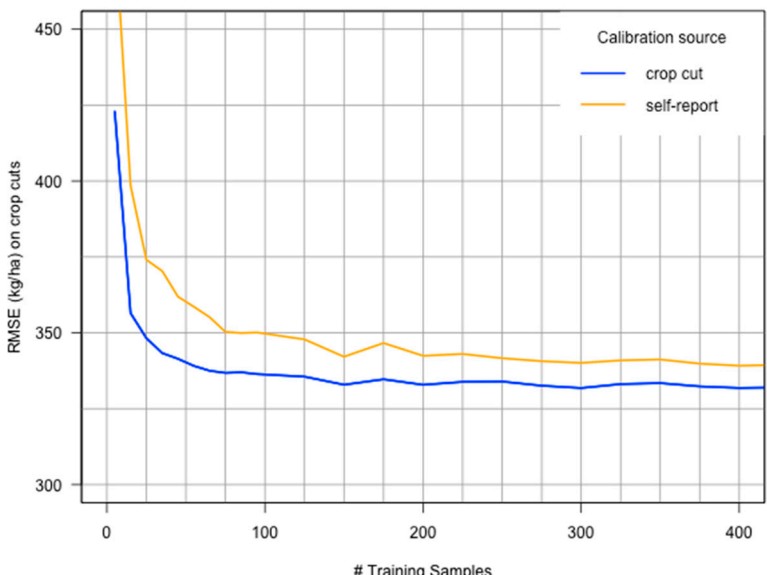

**Figure 6.** The average root mean squared error (RMSE) between crop cut yields and satellite-based yield estimates for fields not used in calibration (i.e., out-of-sample), plotted for different size of training datasets used in calibration and for different sources of yield data for calibration (crop cut or farmer self-report). Values shown are the mean across 200 different random subset of calibration points for each calibration size. Performance improves rapidly with each additional sample until ~30 training points, after which improvement is more gradual.

A final consideration for this study was whether the spatial resolution of the satellite data is a major factor determining the performance of satellite-based models. The Planetscope imagery was processed in the same manner as Sentinel-2, with recursive harmonics fit to the GCVI observations, and the peak of the harmonic fit used to predict yield. We also considered whether combining Sentinel-2 and Planetscope observations, which results in a denser time series of VI observations, led to any

improvements. Overall, the two sensors performed similarly (Figure 7), suggesting that the benefits of a finer resolution (for Planetscope) or better sensor spectral resolution and signal-to-noise (for Sentinel-2) were not substantial in this particular system.

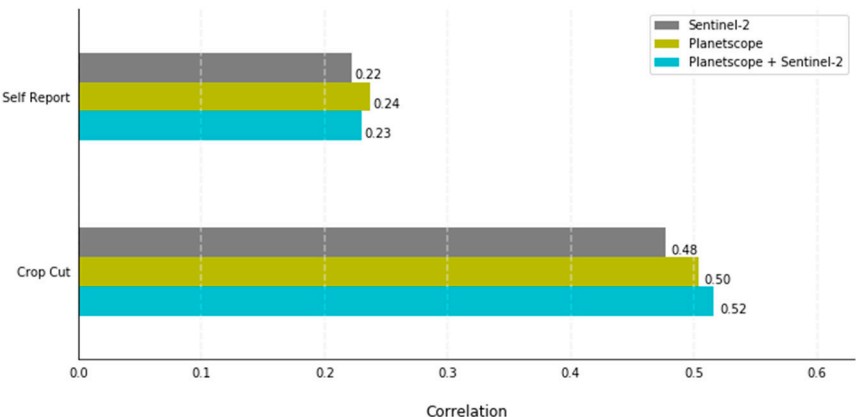

**Figure 7.** Correlation between ground-based yields and satellite peak GCVI for Sentinel-2 (S2), Planetscope (PS), and the combination of the two. Peak GCVI values were determined from a harmonic fit (Equation (5)) to the individual observations of GCVI throughout the season. Performance was similar for the different satellite sensors.

The DG imagery could not be processed in a similar way given the sporadic coverage of fields throughout the season. That is, the differing frequency of DG images would complicate the interpretation of the resulting harmonic coefficients relative to those from Sentinel-2. Therefore, to compare with Sentinel-2 we calculated the Sentinel-2 GCVI from the harmonic regression for the date of the DG image and computed the correlation between GCVI for each sensor and crop cut yields for the sub-set of fields that were located within the image. A comparison of the correlations for the two sensors with crop cut yields revealed little systematic difference between the two sensors (Figure 8). Sentinel-2 outperformed DG in slightly more than half of the image pairs, but differences were typically only a few percentage points.

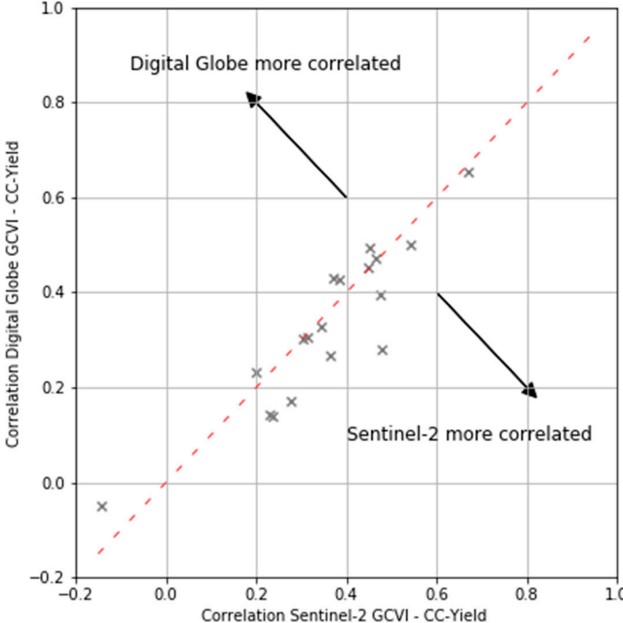

**Figure 8.** Comparison of Digital Globe Worldview and Sentinel 2 GCVI-yield correlations on closely matched dates, using crop cut yields for all plots that fell within the extent of the relevant images.

## 5. Conclusions

The rapidly growing availability of satellite data offers great promise for monitoring smallholder agricultural plots. The results of this study indicate that the perceived performance of satellite-based measures can depend to a large extent on how the performance is measured. In our setting, when compared to ground measures of yield, satellite data agreed much more strongly with crop cuts from $8 \times 8$ m sub-plot than with farmer self-reported yields for the entire plot. Studies that rely solely on comparisons with self-reported yields may therefore understate the ability of satellites to measure yield variation, particularly in subsistence systems where farmers do not typically measure production.

Moreover, even comparisons with crop cuts may not give a complete view of the performance of (crop cut) calibrated satellite yields, given high yield heterogeneity in many fields and the fact that satellite pixels will inevitably contain some signal from outside the crop cut sub-plot. Therefore, we recommend (i) following a regression-based approach to estimate the relationships between yields and factors that are likely to influence yields, including inputs, soil conditions, and management practices, and (ii) investigating the sensitivity of these estimates across different yield measures, as an additional form of evaluation of calibrated satellite-based estimates. In the current study, these factors explained more than twice as much variation in satellite-based yield estimates as for either crop cuts or self-report yields. This finding provides an additional line of evidence that satellite-based yields are no worse, and possibly better, than traditional ground-based measures for assessing yield variation. Of course, obtaining accurate ground-based data on agricultural systems remains important, both for the calibration and evaluation of satellite estimates, as well as for the many other aspects of interest that cannot be measured remotely (e.g., off-farm income). However, measuring production is often one of the most time-intensive and therefore expensive parts of field surveys, and satellite-based estimates can help to reduce the number of fields for which ground-based measures are needed. Our estimates suggest that, at least in this setting, only a few dozen high-quality ground observations would be needed to train an accurate model.

As the use of satellite data becomes more mainstream for monitoring and studying agricultural systems, the relative benefits and costs of different sensors is of increasing relevance. Here we considered three sources of satellite imagery, the public and freely available 10 m data from Sentinel-2 as well as finer resolution imagery from two private sector providers (Planet and DigitalGlobe). For the study setting, where plot sizes averaged 1.5 ha and there were frequent cloud-free Sentinel-2 observations during the growing season, there appeared little benefit to using the finer resolution data. In regions with more cloud cover or smaller fields, the benefits of these other sensors would likely be larger, and more comparisons are needed to understand the conditions under which these sensors provide the most value.

**Author Contributions:** Conceptualization, D.B.L., M.B., T.K.; data processing, S.D.T., C.Y., I.Y.D.; analysis, D.B.L., S.D.T., C.Y., I.Y.D.; writing, D.B.L., M.B., T.K. All authors have read and agreed to the published version of the manuscript.

**Funding:** This research was funded in part by the Global Innovation Fund, and by the Mali Mission of the United States Agency for International Development in support of the broader Living Standards Measurement Study—Integrated Surveys on Agriculture (LSMS-ISA) work program in Mali.

**Acknowledgments:** We thank the ICRISAT-Mali management and field staff for their hard work and their commitment to the success of our methodological experiment. We want to thank particularly at ICRISAT Pierre-Sibiry Traoré for his contribution in the geo-delineation of the study area, Sondo Eloi Somtinda of the African Development Bank for CAPI support, Jourdain Lokossou, Andree Nenkam and Berenger Djoumessi Tiague for excellent fieldwork supervision and data cleaning.

**Conflicts of Interest:** The authors declare no conflict of interest.

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
