# Peer review of "Sight for Sorghums: Comparisons of Satellite- and Ground-Based Sorghum Yield Estimates in Mali"

_remotesensing, doi:10.3390/rs12010100_

Round 1

Reviewer 1 Report

The manuscript is well written, and can be easy to follow. The objectives, results and discussions are clearly supported by the collected data and documents. It may be accepted after correcting the formats of citations.

Author Response

Reviewer 1:

The manuscript is well written, and can be easy to follow. The objectives, results and discussions are clearly supported by the collected data and documents. It may be accepted after correcting the formats of citations.

We thank the reviewer for the encouraging comments. We have checked the format of citations to ensure they are consistent with journal requirements. Some of them had not initially been converted to [#] format, but we think they are all now properly cited.

Reviewer 2 Report

The manuscript is worthy of publication after minor edits. The most serious problem is the authors make statements that seem beyond the data or without reference to sources.

Major comments:

There are several phrases that were quite annoying. Perhaps the title, “Sight for sorghums,” is humorous, but I fail to see it. Why would a type of hair cut, “crop cut,” be used to describe sample plots?

The self-reported yields were provided using a number of different units (footnote page 5). Are the unit conversions responsible for part of the variability in self-reported yields? The relatively low variability of the sample plot data indicates that these data provide a good estimate of the overall mean. Same thing happens in the large fields in developed countries. Random plots are used to calculate the field mean. However, random plots don’t describe the spatial variability within a field – different methods of sampling are used.

Citations and references need to be formatted to journal style.

Minor comments:

L 71: Technically, the statement, “Over a decade of research …” is correct but it is highly misleading.  There are about 40 years of research. Are we to conclude research conducted prior to the lead author is worthless?

L 84-85: Previous sentence concludes with a clause, “but not well studied.” The next sentence is about sorghum being a difficult crop, to study??

L 99-101: No need to outline a scientific paper in the standard format. Please delete.

L 115-117: purestand is an adjective for field, not an adjective for households.

L 135: The accuracy of the eTrex 30 without WAAS may be too poor for the satellite sensors used.

L 154: I think you mean a rich dataset, not rich households.

L 221-222 and elsewhere: why do all of these short equations require 2-3 lines of text?

Author Response

Reviewer 2:

The manuscript is worthy of publication after minor edits. The most serious problem is the authors make statements that seem beyond the data or without reference to sources.

We thank the reviewer for the encouraging comments.

Major comments:

There are several phrases that were quite annoying. Perhaps the title, “Sight for sorghums,” is humorous, but I fail to see it. Why would a type of hair cut, “crop cut,” be used to describe sample plots?

We are open to suggestions on the title but have opted to keep it since we prefer to have a shorthand title to go along with the more descriptive “Comparisons of satellite- and ground-based sorghum yield estimates in Mali.” For “crop cut,” this is the common term for taking samples of harvest within a field, and we have clarified this in the text:

“The second common way to study productivity, typically referred to as a crop cut, is to measure the grain weight harvested from a randomly selected portion of a farmers’ plot (Fermont and Benson 2011).”

The self-reported yields were provided using a number of different units (footnote page 5). Are the unit conversions responsible for part of the variability in self-reported yields? The relatively low variability of the sample plot data indicates that these data provide a good estimate of the overall mean. Same thing happens in the large fields in developed countries. Random plots are used to calculate the field mean. However, random plots don’t describe the spatial variability within a field – different methods of sampling are used.

Good question. Regressing the self-reported yields on the units factors, we find that they explain only 4% of the variability. The unit conversions are the village-level median farmer-reported values. We computed the conversion factors at the village-level in part to reduce potential biases in farmer-reported conversions, while recognizing the possibility of spatial variation in conversion factors based on prior work (Oseni et al., 2017). This information has been added in section 2.1:

“We computed the conversion factors at the village-level in part to reduce potential biases in farmer-reported conversions, while recognizing the possibility of spatial variation in conversion factors based on prior work [14]. In the current study, only 4% of observed variability in self-report yields were explained by the conversion factors used.”

Citations and references need to be formatted to journal style.

We have used the “remote sensing” style available in Mendeley reference manager and checked that all citations are correct.

Minor comments:

L 71: Technically, the statement, “Over a decade of research …” is correct but it is highly misleading.  There are about 40 years of research. Are we to conclude research conducted prior to the lead author is worthless?

Good point, we obviously did not intend to suggest that, and initially had referenced a paper from the 90s which is more than 20 years old (and not by the authors). We have modified the sentence:

“Several decades of research has focused on developing and testing algorithms to estimate yields from satellite, initially in large commercial plots [10,11] and increasingly in smallholder systems [8,9,12,13].”

L 84-85: Previous sentence concludes with a clause, “but not well studied.” The next sentence is about sorghum being a difficult crop, to study??

We have reworded for clarity:

“First, we focus on sorghum, one of the primary staples in sub-Saharan Africa but less commonly studied compared to other staples such as maize and rice. Sorghum is a relatively difficult crop for remote yield estimation given the high variability within and between plots in the cultivars grown by farmers, and the relatively high variation in the harvest index (ratio of grain to total crop biomass) compared to other crops [9].”

L 99-101: No need to outline a scientific paper in the standard format. Please delete.

Done

L 115-117: purestand is an adjective for field, not an adjective for households.

Corrected.

L 135: The accuracy of the eTrex 30 without WAAS may be too poor for the satellite sensors used.

In our case, the eTrex30 instruments are WAAS-enabled, and visual inspection of the polygons indicated that errors appeared sufficiently small to get reliable plot corners. We have now mentioned in the text that they were WAAS-enabled.

L 154: I think you mean a rich dataset, not rich households.

Yes, thanks. We have removed the adjective to avoid confusion.

L 221-222 and elsewhere: why do all of these short equations require 2-3 lines of text?

We were simply trying to make it easily legible. We defer to the copyeditor on the correct final spacing.

Reviewer 3 Report

This manuscript evaluates the performance of optical Satellite data for resolving Sorghum crop yields at small-holder agricultural systems based on uncertain ground measurements. The authors present a detailed analysis of the Earth observation data, including the use of a fitted function to essentially gap-fill observations on cloudy days and the comparison of results when retrieving estimates from sensors at different spatial resolutions. A very thorough quantification of errors in the associated ground measurements (i.e. self-reported vs. crop cut yields) is also presented.

Overall, I found this manuscript very well written and I would recommend accepting this for publication once the small number of relatively minor points are addressed by the authors.

1)  Lines 42-43: It would be good to have some definition of a “smallholder agricultural system” for readers that might not be familiar with the concept.

2) Lines 69-71: I know this is partly addressed later in the manuscript, but it would be good to have some idea about the typical size of a smallholder plot.

3) Lines 139-143: Please consider revising this sentence as it is not clear.

4) Lines 182-185: It would be useful to include the equations for GCVI, NDVI and MTCI here.

5) Lines 235-237: I am not clear on why the optimisation of the harmonic curve fitting would target the upper limits of the curve. Would it not be better to remove the values that show drops in values (i.e. cloud cover days) then apple the fit to the remaining days?

6) Line 375-376: Please add more detail to the Table 3 caption.

7) Line 421: Please define “DG” (i.e. Digital Globe).

Author Response

Reviewer 3:

This manuscript evaluates the performance of optical Satellite data for resolving Sorghum crop yields at small-holder agricultural systems based on uncertain ground measurements. The authors present a detailed analysis of the Earth observation data, including the use of a fitted function to essentially gap-fill observations on cloudy days and the comparison of results when retrieving estimates from sensors at different spatial resolutions. A very thorough quantification of errors in the associated ground measurements (i.e. self-reported vs. crop cut yields) is also presented.

Overall, I found this manuscript very well written and I would recommend accepting this for publication once the small number of relatively minor points are addressed by the authors.

We appreciate the encouraging comments.

1)  Lines 42-43: It would be good to have some definition of a “smallholder agricultural system” for readers that might not be familiar with the concept.

Good point. These are typically defined as under 2ha. We have noted this:

“Despite its simplicity relative to other measures of productivity, accurately measuring crop yields presents a challenge in smallholder agricultural systems, where plots are typically less than 2 ha in size. “

2) Lines 69-71: I know this is partly addressed later in the manuscript, but it would be good to have some idea about the typical size of a smallholder plot.

See above.

3) Lines 139-143: Please consider revising this sentence as it is not clear.

It has been revised:

“The approach to random placement of the crop cut sub-plots, supervision of the crop cut sub-plots throughout the season, and harvesting, processing and tracking of sorghum cultivated in each quadrant were all identical to the approach in an earlier methodological study focused on maize in Eastern Uganda [14].”

4) Lines 182-185: It would be useful to include the equations for GCVI, NDVI and MTCI here.

We have added these as equations (1-3) and updated all other equation numbers and references.

5) Lines 235-237: I am not clear on why the optimisation of the harmonic curve fitting would target the upper limits of the curve. Would it not be better to remove the values that show drops in values (i.e. cloud cover days) then apple the fit to the remaining days?

Good question. The main reason is that we don’t have a robust way of masking clouds, and thus if knowing which values to drop. The recursive fit effectively identifies anomalously low values and drops them. As described in section 3.1: “If the input value was lower, it was replaced with the predicted value, and the resulting time series was then used to refit equation (4)”

6) Line 375-376: Please add more detail to the Table 3 caption.

We have expanded the caption to explain all the symbols:

“Regression results using three alternative measures of sorghum yield: (i) Self-reported, (ii) crop cut, and (iii) crop cut-calibrated Sentinel-2 satellite yields. In first column, carrots (^) indicate inputs that were input as a binary variable (0,1) and asterisk (*) indicate variables that were calculated on a per ha basis using self-report area.  Values in parentheses show the standard error of the estimate, and significance of the coefficient is indicated as (* p <0.1, ** p<0.05, *** p < 0.01).”

7) Line 421: Please define “DG” (i.e. Digital Globe).

We have now defined this at first mention of Digital Globe.